# Adaptation and Validation of the Constructivist Teaching Practices Inventory in Elementary Physical Education (CTPI-EPE) for Brazilian Physical Education Pre-Service Teachers

**DOI:** 10.3390/ijerph191912091

**Published:** 2022-09-24

**Authors:** Ana F. Backes, Valmor Ramos, Ricardo T. Quinaud, Vinicius Z. Brasil, Humberto M. Carvalho, Sergio J. Ibáñez, Juarez V. Nascimento

**Affiliations:** 1Center of Sports, Federal University of Santa Catarina, 88040-900 Florianópolis, SC, Brazil; 2Health and Sport Science Center, State University of Santa Catarina, 88080-350 Florianópolis, SC, Brazil; 3Physical Education Department, University of the Extreme South of Santa Catarina, 88806-000 Criciúma, SC, Brazil; 4Faculty of Sports Sciences, University of Extremadura, 10003 Cáceres, Spain

**Keywords:** psychometric properties, methodology, constructivism, teaching, physical education

## Abstract

Constructivism is used as a powerful theoretical outlook to support teaching, learning, and curriculum in physical education and sport. The Constructivist Teaching Practices Inventory in Elementary Physical Education (CTPI-EPE) is a valid instrument for assessing in-service teachers. However, there is a need to translate constructivist teaching practices for PE into other languages. This study examined the validity of the Portuguese version of the adapted CTPI-EPE for Brazilian physical education preservice teachers (PST). The sample comprised of 869 physical education PSTs from Brazil. Data were collected through an online form. Aiken’s V was used to examine content validity, and Bayesian methods used for exploratory and confirmatory factor analysis. The results showed adequate content and internal structure of the translated and adapted questionnaire. This study highlighted the validity of the Portuguese version of the CTPI-EPE, which could be considered an important instrument for self-reflection by PSTs, and provide information for improved training in higher education toward constructivist teaching.

## 1. Introduction

Constructivist learning theories are used as a theoretical perspective supporting teaching, learning, and curriculum in physical education and sport [1,2,3,4,5]. Constructivist principles provide conceptual foundations for redefining teachers’ roles and establishing new teaching practices. For example, it can be applied in contexts where the activity focuses on game understanding and the cognitive and social engagement of the learners in physical education and sports [6,7,8]. Observations considering teacher education across theoretical perspectives have robustly noted that preservice teachers’ (PST) prior knowledge and experience influence their learning both positively and negatively [4]. Constructivist research that examines the impact of prior knowledge is relevant in identifying pre-existing conceptions that are difficult to change, new concepts that are difficult to learn, the role of pre-existing knowledge in the learning process, and factors that facilitate or constrain learning [9].

Recent observations in PSTs’ learning and the implementation of student-centered models in physical education noted the low impact of initial training on changing PSTs’ beliefs concerning constructivist-oriented practices [10,11]. Physical education PSTs start education programs with personal, preconceived knowledge regarding teaching and the nature of learning from their previous accumulated experience as students and athletes [12,13,14,15,16,17]. In addition, constructivist principles often conflict with naive teaching perspectives [18,19], given that PSTs can interpret learning as a transmissive process, during which PE teachers should rely on the reproduction of knowledge, and the teacher’s authority plays a determinant role in the learning process [19]. Therefore, conceptual change is a difficult, long-term process [9]. Physical Education Teacher Education (PETE) programs should support this conceptual change and help PSTs to reconceptualize teaching and learning in physical education [16,18,19].

There are few instruments available with which to examine the perceptions of PSTs regarding constructivist teaching practices [20,21,22] and behaviors [23]. The Constructivist Teaching Practices Inventory in Elementary Physical Education (CTPI-EPE) was originally developed and validated in order to identify the constructivist-oriented teaching practices of teachers in the United States [8]. The questionnaire has since been validated in other contexts, such as Turkey [24], Greece [25], and Brazil [26]. However, in the Brazilian context, only content validity was performed. CTPI-EPE has been noted as a valid instrument by which to measure in-service teachers [8,24]. Data based on the questionnaire has provided specific direction for curricular and instructional approaches that successfully improve PSTs’ learning regarding constructivist-oriented teaching practices [8]. Nevertheless, there is no information available concerning the cross-cultural validity of the questionnaire when applied to physical education PSTs’ constructivist-oriented teaching practices, or its use in exploring sources of variation across PSTs’ constructivist-oriented teaching practices, as was recommended during original conception [8].

Lastly, the psychometric properties of questionnaires are often examined using frequentist methods, often leading to limited or inaccurate interpretations of the data [27]. On the other hand, Bayesian methods consider the prior information available and the information contained in the data to update knowledge [27]. In this study, we examined the validity of the adapted version of the CTPI-EPE to Portuguese in Brazilian physical education PSTs.

## 2. Materials and Methods

### 2.1. Study Design

The research ethics committee of the Santa Catarina State University, Brazil, approved the present research with ID 4.802.198/2021. In this study, we accessed evidence based on test content and internal structure of the Portuguese version of the CTPI-EPE, following APA standards [28].

### 2.2. Participants

Content validity was carried out using an ‘expert judgment method’ [29]. The expert cohort comprised of 13 PhDs in physical education, currently working in Brazilian public higher education institutions, and with at least ten years of professional experience. For the internal structure analysis, a sample comprised of 869 (female = 399 and male = 470) Brazilian physical education PSTs from all five regions of Brazil (North = 36, Northeast = 72, Midwest = 53, Southeast = 74, and South = 634). Participants considered in this study were PSTs regularly enrolled in Brazilian public and private higher education institutions. The “RAND” function in Microsoft Excel 365 (Microsoft Corporation, version 16.65) was used to generate a unique random number. The dataset was reordered from lowest to highest numbers, the first half of the dataset was used to run Bayesian Exploratory Factor Analysis (BEFA), and the second half was used to run Bayesian Confirmatory Factor Analysis (BCFA).

### 2.3. Instrument and Procedures

We used a translated and adapted version of the CTPI-EPE, which was initially developed and validated in English in the United States [8]. The CTPI-EPE comprises of 36 items, distributed in four dimensions: Facilitating Active Construction of Knowledge in Dance and Gymnastics; Facilitating Active Construction of Knowledge in Games and Skills; Facilitating Personal Relevance; Facilitating Social Cooperation. The items in the first and second dimensions refer to teaching situations that promote students’ responsibility for their learning, involving them in interesting and challenging activities of discovery, self-regulation, and problem-solving in dance and gymnastics (dimension one) and games and skills (dimension two). On the other hand, the items of the third dimension are related to the creation of teaching situations that consider the previous knowledge and experiences of the students in the organization of learning activities and the presentation of contents. Finally, items belonging to the fourth dimension are linked to creating opportunities for students to share ideas and solve learning problems together [8].

The original version of the CTPI-EPE went through an adaptation process in which items from the dimension ‘Facilitating Active Construction of Knowledge in Dance and Gymnastics’ (n = 11) were excluded. The exclusion decision arose because dance and gymnastics have little influence on Brazilian culture and physical education classes compared to games and sports. Games are a significant component of the physical education curriculum, approximately 65% or more of time is allotted to games and sports [30]. Furthermore, there is evidence that constructivist teaching practices benefit sports learning [31]. Moreover, the Turkish version also excluded this dimension in their validation process due to cultural features [24]. Therefore, the initial Portuguese version of the CTPI-EPE comprises of 25 items distributed in three dimensions: ‘Facilitating Active Construction of Knowledge in Games and Skills’ (9 items), ‘Facilitating Personal Relevance’ (10 items), and ‘Facilitating Social Cooperation’ (6 items), measured by a five-point intervals scale (‘never’ to ‘always’).

Based on the 25-item structure, two native Portuguese speakers (a certified specialist in English-Portuguese translation and a PhD professor in physical education) produced independent translations of the CTPI-EPE [32,33]. First, both versions were compared and a synthesis of the document was prepared. Second, an English reviewer conducted a back-translation following a blind translation procedure. Third, the original and back-translated versions were compared to eliminate translation errors.

Initially, 13 experts served as judges in evaluating the clarity of language, practical pertinence, and theoretical dimensions of the Portuguese version of the CTPI-EPE, and could also provide suggestions to improve the instrument quality. Then, after content validity, the Portuguese version of the CTPI-EPE was administered to the Brazilian sample of PSTs through an online form. Participants were informed about the study’s purpose and assured of confidentiality.

### 2.4. Data Analysis

We verified the Portuguese version of the CTPI-EPE content validity index using Aiken’s V [34] and made use of the Visual Basic program [35], which is a tool commonly used in studies for sport science [36,37,38]. The validity index can variate between 0 to 1, and if the index is closer to 1, it means a higher correlation. The minimum value is determined by the number of instrument items and responses interval. The formula proposed by Aiken [34] was used to establish the criteria for modifying or eliminating items. The cut-off point set for excluding an item was 0.68.

We explored the psychometric properties of the questionnaire using BEFA and BCFA. To access evidence based on internal structure, a BEFA was conducted first. Based on the original version of the questionnaire [8] and the exclusion of a full dimension, we constrained the factorial structure to three latent factors maximum (Kmax). To conduct this analysis, we ran a total of 100,000 interactions. The minimum posterior mean for retaining an item was set as 3, although there was no recommendation in the literature. The value was established for closer interpretation of traditional exploratory factor analysis (frequentist value of “0.3”). Items with posterior means lower than 3 could be excluded from conducting the subsequent analysis. Metropolis-Hastings’ acceptance rate was used to retain the posterior probabilities of items being different from zero, and a default identification restriction (Nid = 1) was used to identify the number of manifest variables to each factor. The analysis was conducted using the package “BayesFM”, available in the R statistical language [39]. Next, we ran a BCFA with the resulting psychometric structure. We set the posterior latent variables closer to 0.5 as acceptable for retaining an item [40]. We used several fit indexes to confirm the model fit (Bayesian root mean square error of approximation—BRMSEA, Bayesian Gamma Hat—BGammaHat, Adjusted Bayesian Gamma Hat—adjBgammahat, and Bayesian McDonald’s centrality index—BMc) [41].

Furthermore, we used fit measures (e.g., npar, waic, bic, looic) to compare and find the best model. Lastly, we checked the items’ standardized residuals covariances. Items with standardized residuals covariances closer to 0.1 or −0.1 were excluded. We ran the models with two chains for 8000 iterations, with 2000 used as a warm-up. We applied normal prior (0, 10) for the manifest variable (intercept) and normal prior (0, 1) for the latent variable to regularize the models. Finally, we assessed reliability by Cronbach Alpha coefficient. Values greater than 0.70 were considered indicators of suitable reliability [29].

## 3. Results

The Portuguese version of the CTPI-EPE presented the content validity indexes of general Aiken *V* (*V* = 0.94), clarity of language (*V* = 0.91), practical relevance (*V* = 0.96), and theoretical dimension (*V* = 0.96) (Table 1).

BEFA showed that a 3-factor structure best fitted the questionnaire structure. All 25 items had posterior means higher than 3 (Table 2). Factor 1 (Facilitating Active Construction of Knowledge in Games and Skills) is composed of five items (1, 4, 5, 7, and 8), Factor 2 (Facilitating Personal Relevance) is composed of seven items (9, 10, 11, 12, 14, 15, and 16), and Factor 3 (Facilitating Social Cooperation) is composed of 13 items (17, 18, 20, 21, 22, 23, 24, 25, 26, 28, 32, 33, and 36).

Based on the construct evidence from BEFA, we tested the factor structure in the BCFA. The first model (3-factor structure with 25 items) presented adequacy of the posterior values (>0.5; Table 3), but inadequate fit indexes (Table 4; BRMSEA = 0.08; BGammaHat = 0.88; adjBgammahat = 0.85; BMc = 0.44). Additionally, we tested items standardized residuals covariances (Appendix A).

Based on this analysis, we found that some items presented high-standardized residuals covariances that could disturb model fit. Thus, items 10, 14, 17, 20, 24, and 33 were excluded. The second model (3-factor structure with 19 items) was run. Model 2 presented satisfactory posterior values (>0.5), better fit measures than Model 1 (meaning a better model than Model 1) and adequate fit indexes (BRMSEA = 0.06; BGammaHat = 0.94; adjBgammahat = 0.91; BMc = 0.74). However, items 5 and 23 presented high-standardized residuals covariances (>0.1). Although model fit presented satisfactory values, we decided to exclude item 23 to test a third model, and to avoid items from different factors with high-standardized residuals covariances.

The third model (3-factor structure with 18 items) presented satisfactory factor loadings, better fit measures, and similar fit indexes (BRMSEA = 0.06; BGammaHat = 0.94; adjBgammahat = 0.92; BMc = 0.77) than Model 2. Additionally, items did not present standardized residuals covariances closer to 0.1 or −0.1. Model fit measures are presented in Table 5. Thus, Model 3 was considered the final model (Figure 1). The final model showed good internal consistency across the three dimensions (FACK = 0.86; FPR = 0.90; FSC = 0.94). The structure is composed of Factor 1 (Facilitating Active Construction Knowledge in Games and Skills) with five items (1, 4, 5, 7, and 8), Factor 2 (Facilitating Personal Relevance) with five items (9, 11, 12, 15, and 16) and Factor 3 (Facilitating Social Cooperation) with eight items (18, 21, 22, 25, 26, 28, 32, and 36). The numbering of the 18 items was ordered for the final version (Appendix B).

## 4. Discussion

This study examined the validity of the adapted version of the CTPI-EPE to Portuguese. The analytical framework was important to help sustain the structure of the questionnaire, and demonstrated adequate psychometric proprieties making it shorter and easier to apply. As a result, the Portuguese version of the CTPI-EPE presented good content validity (*V* = 0.94), considered very high [34] even compared to other studies in sports science [37,38,42] and the initial content validity version in Brazil [26].

Based on content validity, a three-factor structure was deemed the best fit, and all 25 items presented posterior means higher than our cut-off point [8]. Furthermore, the three-factor structure (Facilitating Active Construction of Knowledge; Facilitating Personal Relevance; Facilitating Social Cooperation) is theoretically supported by the three fundamental constructivist principles [4,8], consistent with the Turkish version [24]. Additionally, our results are consistent with observations made using an observational instrument that examines constructivist teaching practices in physical education research [23].

Regarding the distribution of items in the factors, three items (5, 7, and 8) originally from the dimension “Facilitating Personal Relevance” migrated to the dimension “Facilitating Active Construction of Knowledge”. The changes in the items emphasize the importance of learners’ active and constructive involvement in their learning.

These observations highlight the importance of providing the opportunity for students to develop responsibility for their learning, actively involving them in discovery, self-regulation, problem-solving, and encouraging them to be autonomous thinkers and evaluate their learning [4,7,8]. In the dimension “Facilitating Personal Relevance” (Factor 2), only item 12 from Factor 1 was included in the model. It refers to teaching situations that establish a relationship between the learner and the content, considers their previous knowledge of organizing learning activities [4,8], and achieves personal relevance and the attribution of meaning to the game in their lives [43]. Although there are differences between the dimensions, both dimensions are related to psychological constructs and are explained by the cognitive constructivist perspective of learning. For example, the Facilitating Active Knowledge dimension develops group strategies that encourage students to take responsibility for their learning, and the Facilitating Personal Relevance dimension addresses strategies to integrate students’ new understanding with their previous knowledge [4,8]. This is an important aspect to consider when interpreting the differences between these dimensions.

In the “Facilitating Social Cooperation” dimension (Factor 3), all items (20, 21, 22, 23, 24, and 25) from the original version remained. In addition, six items (17, 18, 26, 28, 32, and 36) from Factor 1 and one item (33) from Factor 2 were included in this dimension. Our observations contrast with both the original and the Turkish version. It is likely that Brazilian physical education PSTs perceive that providing opportunities for student involvement in games are strategies that encourage active “participation”. These strategies involve learning and social interaction with others (peers and teachers), related to the sociocultural constructivism perspective [3,5,8]. Through interaction, students share ideas, expose common limitations, establish comparisons, negotiate meanings, ask questions that contribute to individual understanding, make decisions together, listen, and carefully observe each other for mutual group adjustment [2,44]. Thus, there is strong empirical support for the social domain as an essential pillar of learning, as long as all the students demonstrate competence and active learning throughout the process [45].

BCFA confirmed the three factors had good fit indices [40,41]. However, seven items (10, 14, 17, 20, 23, 24, and 33) were excluded. Items 14, 20, and 33 displayed cross-loading in the original version, subjectively included in the original questionnaire in the respective dimensions [8]. Additionally, the subjective inclusion of these items in the original version might have supported the maintenance of many controversial items. Thus, excluding them did not have any effect on the final model and its fit indexes. The exclusion of items 17, 23, and 24 can be justified by interpreting the Brazilian PSTs’ active involvement as similar to social cooperation concepts previously discussed. Additionally, these items were excluded in the confirmatory factor analysis of the Turkish version [24]. Brazilian PSTs showed similar perceptions regarding items 9 and 10, as both items referred to students’ understanding and previous experiences of games and skills, and their relationship with learning new content [4,8], resulting in the exclusion of item 10.

Finally, the Portuguese version of the CTPI-EPE obtained a final structure with 18 items distributed across three factors. The distribution and number of items diverged from the Turkish version, with 16 items [24]. It is important to highlight that in addition to cultural differences, the present Brazilian version is validated for preservice teachers. In contrast, the original version [8], Turkish version [24], and versions used in Greece [25] and Brazil [26], apply to in-service teachers. Limitations of the study were related to the majority of the sample consisting of physical education preservice teachers from public universities in southern Brazil. A more balanced distribution of the sample in different regions of Brazil, and validation in other cultures of Portuguese speakers (e.g., Brazil and Portugal), is recommended for future studies.

## 5. Conclusions

Conditional on the data, the Portuguese version of the CTPI-EPE is a valid and reliable instrument, providing a valuable understanding of PSTs living in Portuguese-speaking countries. Furthermore, the questionnaire presented satisfactory evidence based on test content and internal structure, indicating validity for assessing Brazilian physical education PSTs’ constructivist-oriented teaching practices. Thus, CTPI-EPE can serve as a self-reflective tool for PSTs to identify and implement constructivist teaching practices. Hence, the information generated from the Portuguese version of the CTPI-EPE may provide insight into the development of the preservice teacher curriculum in PETE.

## Figures and Tables

**Figure 1 ijerph-19-12091-f001:**
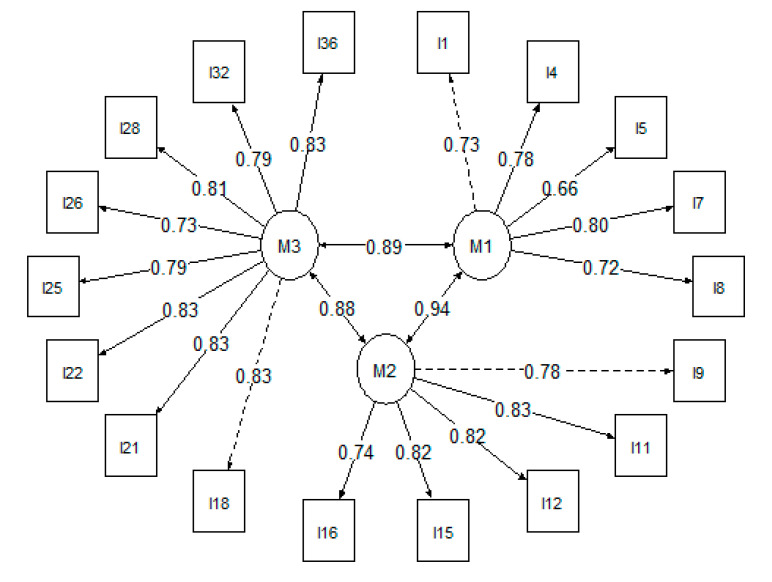
The final model of the CTPI-EPE. Note. M1 = Facilitating Active Construction Knowledge in Games and Skills; M2 = Facilitating Personal Relevance; M3 = Facilitating Social Cooperation; I = Item.

**Table 1 ijerph-19-12091-t001:** Aiken`s *V* index.

	Clarity of Language			Practical Relevance			Theoretical Dimension			
		IC 95%			IC 95%			IC 95%		*V* Total
Item	*V*	*Min*	*Max*	*V*	*Min*	*Max*	*V*	*Min*	*Max*	
1	0.87	0.75	0.93	0.95	0.84	0.98	0.94	0.84	0.98	0.92
4	0.85	0.72	0.92	0.91	0.80	0.96	0.94	0.84	0.98	0.90
5	0.79	0.66	0.88	0.89	0.77	0.95	0.91	0.80	0.96	0.86
7	0.92	0.82	0.97	1.00	0.93	1.00	0.98	0.90	1.00	0.97
8	0.89	0.77	0.95	0.92	0.82	0.97	0.94	0.84	0.98	0.92
9	0.87	0.75	0.93	0.95	0.84	0.98	0.91	0.80	0.96	0.89
10	0.89	0.77	0.95	0.95	0.84	0.98	0.94	0.84	0.98	0.93
11	0.85	0.72	0.92	0.91	0.80	0.96	0.91	0.80	0.96	0.89
12	0.94	0.84	0.98	0.96	0.87	0.99	0.98	0.90	1.00	0.96
14	0.91	0.80	0.96	0.96	0.87	0.99	0.96	0.87	0.99	0.94
15	0.96	0.87	0.99	0.98	0.90	1.00	0.96	0.87	0.99	0.96
16	0.91	0.80	0.96	0.94	0.84	0.98	0.98	0.90	1.00	0.94
17	1.00	0.93	1.00	1.00	0.93	1.00	1.00	0.93	1.00	1.00
18	0.96	0.87	0.99	0.96	0.87	0.99	0.98	0.90	1.00	0.97
20	0.92	0.82	0.97	0.94	0.84	0.98	0.94	0.84	0.98	0.93
21	0.79	0.66	0.88	0.85	0.72	0.92	0.89	0.77	0.95	0.84
22	0.83	0.70	0.91	0.98	0.90	1.00	0.98	0.90	1.00	0.90
23	1.00	0.93	1.00	1.00	0.93	1.00	1.00	0.93	1.00	1.00
24	0.96	0.87	0.99	1.00	0.93	1.00	1.00	0.93	1.00	0.98
25	0.94	0.84	0.98	0.89	0.77	0.95	0.94	0.84	0.98	0.94
26	0.92	0.82	0.97	0.94	0.84	0.98	0.94	0.84	0.98	0.93
28	0.92	0.82	0.97	1.00	0.93	1.00	0.98	0.90	1.00	0.95
32	1.00	0.93	1.00	1.00	0.93	1.00	1.00	0.93	1.00	1.00
33	0.96	0.87	0.99	1.00	0.93	1.00	1.00	0.93	1.00	0.98
36	1.00	0.93	1.00	1.00	0.93	1.00	1.00	0.93	1.00	1.00
Total	0.91	0.81	0.96	0.96	0.86	0.98	0.96	0.87	0.99	0.94

**Table 2 ijerph-19-12091-t002:** Bayesian Exploratory Factor Analysis posterior means.

Item	FACK	FPR	FSC
1	3.74		
4	4.20		
5	3.68		
7	4.23		
8	3.82		
9		4.09	
10		4.19	
11		4.17	
12		4.16	
14		3.47	
15		4.12	
16		4.30	
17			3.80
18			3.92
20			4.06
21			4.02
22			4.00
23			4.00
24			4.12
25			4.20
26			3.86
28			4.05
32			4.05
33			4.24
36			4.19

Note. FACK = Facilitating Active Construction Knowledge in Games and Skills; FPR = Facilitating Personal Relevance; FSC = Facilitating Social Cooperation.

**Table 3 ijerph-19-12091-t003:** Bayesian Confirmatory Factor Analysis posterior latent variables.

	M1			M2			M3		
Item	FACK	FPR	FSC	FACK	FPR	FSC	FACK	FPR	FSC
1	0.73			0.73			0.73		
4	0.78			0.78			0.78		
5	0.66			0.66			0.66		
7	0.80			0.80			0.80		
8	0.72			0.72			0.72		
9		0.80			0.78			0.78	
10		0.84			-			-	
11		0.84			0.83			0.83	
12		0.80			0.82			0.82	
14		0.51			-			-	
15		0.82			0.82			0.82	
16		0.74			0.74			0.74	
17			0.69			-			-
18			0.84			0.83			0.83
20			0.87			-			-
21			0.84			0.83			0.94
22			0.82			0.83			0.92
23			0.73			0.72			-
24			0.77			-			-
25			0.80			0.80			0.79
26			0.72			0.73			0.73
28			0.80			0.81			0.81
32			0.79			0.79			0.79
33			0.78			-			-
36			0.82			0.82			0.82

Note. M1 = Model 1; M2 = Model 2; M3 = Model 3; FACK = Facilitating Active Construction Knowledge in Games and Skills; FPR = Facilitating Personal Relevance; FSC = Facilitating Social Cooperation.

**Table 4 ijerph-19-12091-t004:** Bayesian Confirmatory Factor Analysis fit indexes.

Model	BRMSEA	BGammaHat	adjBgammahat	BMc
M1	0.08	0.88	0.85	0.44
M2	0.06	0.94	0.91	0.74
M3	0.06	0.94	0.92	0.77

Note. M1 = Model 1; M2 = Model 2; M3 = Model 3; BRMSEA = Bayesian Root Mean Square Error of Approximation; BGammaHat = Bayesian Gamma Hat; adjBgammahat = Adjusted Bayesian Gamma Hat; BMc = Bayesian McDonald’s Centrality Index.

**Table 5 ijerph-19-12091-t005:** Bayesian Confirmatory Factor Analysis models’ fit measures.

Model	Npar	Looic	Bic	Waic
M1	78,000	25,188.46	25,495.41	25,188.28
M2	60,000	19,353.89	19,590.37	19,353.75
M3	57,000	18,256.61	18,480.94	18,256.45

Note. M1 = Model 1; M2 = Model 2; M3 = Model 3.

## Data Availability

Please note that the data are available upon request from the corresponding author.

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
