# Peer review of "Adaptation and Validation of the Constructivist Teaching Practices Inventory in Elementary Physical Education (CTPI-EPE) for Brazilian Physical Education Pre-Service Teachers"

_ijerph, 2022, doi:10.3390/ijerph191912091_

Round 1
Reviewer 1 Report
This was a well designed, implementd, and expressed research study on the validity of a constructivist teaching measure.
The introduction was well-written but appeared to over-emphasize constructivist teaching as a panacea/cure for all learning challenges and other methods/beliefs as "naive". The literature is clear that there are also merits for less-constructivist methods such as direct teaching depending on the aims and context/setting. I believe the authors need to better situate constructivist teaching as such based on the literature.
You state in Line 237 that: "six items (17, 18, 26, 28, 32, and 36) from factor 1 and one item 237 (33) from factor 2 were included in the dimension" and in the discussion that "3 items migrated from factor/dimenion 2 to factor/dimension 1". Can you better explain these as they are confusing relative to your other results? How did these migreated items alter the actual content validity of the two affected dimensions? Were these item migrations part of the final model?
In addition to the exploratory and confirmatory factor analyses, it seems important and necessary to report the alpha reliability of all three final dimensions.
The discussion is overly comprised of results that should've been reported in the results section. I recommend that you revamp the discussion to highlight only critical or summative results and then discuss those relative to previous research on this measure.
So two of the three dimensons you assessed in this study changed by reducing the quantity of items by 2-5, had 7 items deleted because of cross-loadings, and had several items migrating between dimensions. How was the content validity of these revised and final dimensions altered as a result?
It seems necessary to me that you better justify how this adapted and translated measure is actually a valid measure of constructivist practice especially when I believe that many of the items (listed in your appendix) are also reflective of less constructivist teaching practices such as direct instruction.
Author Response
Dear editor,
Thank you for considering our work suitable for the submission process of the International Journal of Environmental Research and Public Health and also for the reviewer comments. Hopefully, we have addressed each question/comment in a satisfactory manner so that the revision is acceptable for publication in the Journal. Specific responses to the comments/questions of the reviewers are indicated subsequently in BLUE font.
Reviewer #1
English language and style are fine/minor spell check required.
Authors' reply: We revised and checked the English language and style.
This was a well-designed, implemented, and expressed research study on the validity of a constructivist teaching measure.
The introduction was well-written but appeared to over-emphasize constructivist teaching as a panacea/cure for all learning challenges and other methods/beliefs as "naive". The literature is clear that there are also merits for less-constructivist methods such as direct teaching depending on the aims and context/setting. I believe the authors need to better situate constructivist teaching as such based on the literature.
Authors' reply: We thank the reviewer's comments. We agree that other methods also have merit in teaching. Thus, we added information to be more explicit about the specific contexts and goals (e.g., game understanding and learner cognitive and social engagement) that constructivism would be a better method of teaching based on the literature (see L33-35).
You state in Line 237 that: "six items (17, 18, 26, 28, 32, and 36) from factor 1 and one item (33) from factor 2 were included in the dimension" and in the discussion that "3 items migrated from factor/dimenion 2 to factor/dimension 1". Can you better explain these as they are confusing relative to your other results? How did these migreated items alter the actual content validity of the two affected dimensions? Were these item migrations part of the final model?
Authors' reply: The exploratory factor analysis showed that the Facilitating Social Cooperation' dimension loaded 13 items (17, 18, 20, 21, 22, 23, 24, 25, 26, 28, 32, 33, and 36). Thus, all items (20, 21, 22, 23, 24, 25) from the original version remained. In addition, six items (17, 18, 26, 28, 32, and 36) from factor 1 and one item (33) from factor 2 were included in this dimension. Therefore, we clarified the sentence, including the six items that were part of the original instrument (see 252-253). Each instrument dimension's fundamental idea/concept was maintained after the migration of the items. However, we considered and highlighted the influence of pre-service teachers' perception of the items in the discussion. For example, probably Brazilian physical education PSTs perceive that providing opportunities for student involvement in games are strategies that encourage active "participation", which involves learning and social interaction with others (peers and teachers). Therefore, six items were migrated from the dimension facilitating active knowledge to the dimension facilitating social cooperation. Lastly, based on the evidence of the validity of the confirmatory factor analysis, some of these items remained and were considered suitable for the final model. Hopefully, we have addressed all concerns raised in the questions.
In addition to the exploratory and confirmatory factor analyses, it seems important and necessary to report the alpha reliability of all three final dimensions.
Authors' reply: The information was added to methods and results (see L162-164; L206-208).
The discussion is overly comprised of results that should've been reported in the results section. I recommend that you revamp the discussion to highlight only critical or summative results and then discuss those relative to previous research on this measure.
Authors' reply: We appreciate the reviewer's comment. We restructured the results and discussion accordingly. Hopefully, the text is now clear both in the results and discussion.
So two of the three dimensions you assessed in this study changed by reducing the quantity of items by 2-5, had 7 items deleted because of cross-loadings, and had several items migrating between dimensions. How was the content validity of these revised and final dimensions altered as a result?
Authors' reply: It is important to highlight that each instrument's dimension's fundamental idea/concept was maintained after the migration of the items. However, we considered and highlighted the influence of PSTs perception on the items in the discussion. It seems reasonable to infer that Brazilian physical education PSTs perceive that providing opportunities for student involvement in games are strategies that encourage active "participation", which involves learning and social interaction with others (peers and teachers). Therefore, six items were migrated from the dimension facilitating active knowledge to the dimension facilitating social cooperation. It is also necessary to clarify that high-standardized residuals showed that some items appeared to the team as redundant. Additionally, the subjective inclusion of these items in the original version might have supported the maintenance of many controversial items. Thus, excluding them did not affect the final model and its fit indexes. Finally, the study's limitations were included at the end of the discussion (see L282-286).
It seems necessary to me that you better justify how this adapted and translated measure is actually a valid measure of constructivist practice especially when I believe that many of the items (listed in your appendix) are also reflective of less constructivist teaching practices such as direct instruction.
Authors' reply: In the CTPI-EPE adaptation process, only 11 items from the dimension 'Facilitating Active Construction of Knowledge in Dance and Gymnastics' were excluded. The exclusion of these items was the only adaptation performed, and the others items were translated. Furthermore, the construction of the original items is supported by important works in the literature on constructivism and teacher education (e.g., Cochran, DeRutier, & King, 1993; Burry-Stock, 1995) and physical education (Rovegno, 1993). According to the authors: "The major tenet of constructivist-oriented teaching that is, facilitating learners' active construction of knowledge through integration of learners' new understanding with their prior understanding and social interaction with each other served as a theoretical framework for developing the items on the CTPI-EPE" (Chen et al., 2000, p. 28). Finally, the items went through the content validation process with experts in the United States (Chen et al., 2000), Turkey (AÄŸbuÄŸa, 2013), and now in Brazil.
Cochran, K.F., DeRuiter, J.A., & King, R.A. (1993). Pedagogical content knowledge: An integrative model for teacher preparation. Journal of Teacher Education, 44(4), 263-272.
Rovegno, I. (1993). Content-knowledge acquisition during undergraduate teacher education: Overcoming cultural templates and learning through practice. American Educational Research Journal, 30(3), 61-64.
Burry-Stock, J.A. (1995). Expert science teaching evaluation model (ESTEEM): Theory, development, and research. Kalamazoo, MI: The Evaluation Center, Western Michigan University.

Reviewer 2 Report
A few grammar mistakes need to be corrected throughout the paper (For example, in the last paragraph of the of the data analysis portion of the paper, some sentences are in present tense and others are in past tense (pp 138 -160). Re-word them in past tense.
Introduction
General comment: a better case needs to be made for the need to examine (or re-examine) the validity of CTPI-EPE in Portuguese.
- There are a few sentences that need improvement for clarity:
Line 46 - Also, constructivist principles often conflict with the PSTs’ naive teaching perspectives
Line 55 - The questionnaire has been validated and reported in other contexts, such as in Turkey [24], Greece [25], and Brazil [26].
- If the validity of the CTPI-EPE has been previously examined in Brazil, why is this paper doing the same work? It seems that the sentence in line 56 is conflicting with the sentence in line 59. You need to make a better case of what validity evidence has been previously provided (reference 26), what this paper is adding and why it is necessary.
Methods
General comment: The methods in general were very solid: The analysis of content by experts and translation procedures used adequate. Use of two separate samples for the confirmatory and exploratory analyses was an appropriate choice.
Results
General comment: The description of the results was also well done.
Line 177 - Improve the sentence ‘not well fit indexes’
Discussion
General comment: The discussion was clear overall. A brief mention of previous validation work in Brazil and how this paper is adding to that work may help. A little improvement in the English used is necessary in the discussion as well. For example, the first sentence (line 212) could be improved if you switch the order of the questionnaire in the sentence ‘This study examined the validity of the adapted version of the CTPI-EPE to Portuguese.’
- The paragraph between lines 224 and 235 was difficult to read. The wording for the explanation of the migration of items was difficult to follow and need some improvement for better clarity.
- Limitations should be mentioned
Conclusion
- Line 266 (‘Conditional on the data, the Portuguese version of the CTPI-EPE is a valid and reliable instrument, providing a valuable instrument to understand PSTs living in Portuguese- 267 speaking countries’). I do not think you tested the reliability of the instrument in this paper.
Author Response
Dear editor,
Thank you for considering our work suitable for the submission process of the International Journal of Environmental Research and Public Health and also for the reviewer comments. Hopefully, we have addressed each question/comment in a satisfactory manner so that the revision is acceptable for publication in the Journal. Specific responses to the comments/questions of the reviewers are indicated subsequently in BLUE font.
Reviewer #2
A few grammar mistakes need to be corrected throughout the paper (For example, in the last paragraph of the of the data analysis portion of the paper, some sentences are in present tense and others are in past tense (pp 138 -160). Re-word them in past tense.
Authors' reply: We revised the text and hopefully corrected all grammar mistakes.
Introduction - General comment: a better case needs to be made for the need to examine (or re-examine) the validity of CTPI-EPE in Portuguese.
Authors' reply: We agree with the reviewer's comment. We added this information in introduction (see L59-60).
- There are a few sentences that need improvement for clarity: Line 46 - Also, constructivist principles often conflict with the PSTs' naive teaching perspectives.
Authors' reply: We rephrased our argument, also in line with the previous reviewer's previous comment (see L48-50).
- There are a few sentences that need improvement for clarity: Line 55 - The questionnaire has been validated and reported in other contexts, such as in Turkey [24], Greece [25], and Brazil [26].
Authors' reply: We rephrased our argument, and hopefully is now clear.
- If the validity of the CTPI-EPE has been previously examined in Brazil, why is this paper doing the same work? It seems that the sentence in line 56 is conflicting with the sentence in line 59. You need to make a better case of what validity evidence has been previously provided (reference 26), what this paper is adding and why it is necessary.
Authors' reply: We added the information as suggested.
Methods - General comment: The methods in general were very solid: The analysis of content by experts and translation procedures used adequate. Use of two separate samples for the confirmatory and exploratory analyses was an appropriate choice.
Results - General comment: The description of the results was also well done.
Line 177 - Improve the sentence 'not well fit indexes'
Authors' reply: We corrected the sentence.
Discussion - General comment: The discussion was clear overall. A brief mention of previous validation work in Brazil and how this paper is adding to that work may help.
Authors' reply: We added the information as suggested.
A little improvement in the English used is necessary in the discussion as well. For example, the first sentence (line 212) could be improved if you switch the order of the questionnaire in the sentence 'This study examined the validity of the adapted version of the CTPI-EPE to Portuguese.'
Authors' reply: We revised and checked the English language and style.
- The paragraph between lines 224 and 235 was difficult to read. The wording for the explanation of the migration of items was difficult to follow and need some improvement for better clarity.
Authors' reply: We clarified the sentence as suggested.
- Limitations should be mentioned
Authors' reply: We added limitations of the study as suggested.
Conclusion - Line 266 ('Conditional on the data, the Portuguese version of the CTPI-EPE is a valid and reliable instrument, providing a valuable instrument to understand PSTs living in Portuguese-267 speaking countries'). I do not think you tested the reliability of the instrument in this paper.
Authors' reply: We agree with the comment. Hence, we added reliability estimates of all three final dimensions of CTPI-EPE to support our conclusion.

Round 2
Reviewer 1 Report
The authors have adequately addressed each of my concerns in the revision. Good work!